# Marine Anticancer Agents: An Overview with a Particular Focus on Their Chemical Classes

**DOI:** 10.3390/md18120619

**Published:** 2020-12-04

**Authors:** Marilia Barreca, Virginia Spanò, Alessandra Montalbano, Mercedes Cueto, Ana R. Díaz Marrero, Irem Deniz, Ayşegül Erdoğan, Lada Lukić Bilela, Corentin Moulin, Elisabeth Taffin-de-Givenchy, Filippo Spriano, Giuseppe Perale, Mohamed Mehiri, Ana Rotter, Olivier P. Thomas, Paola Barraja, Susana P. Gaudêncio, Francesco Bertoni

**Affiliations:** 1Department of Biological, Chemical and Pharmaceutical Sciences and Technologies (STEBICEF), University of Palermo, 90123 Palermo, Italy; marilia.barreca@unipa.it (M.B.); virginia.spano@unipa.it (V.S.); alessandra.montalbano@unipa.it (A.M.); paola.barraja@unipa.it (P.B.); 2Faculty of Biomedical Sciences, Institute of Oncology Research, USI, 6500 Bellinzona, Switzerland; filippo.spriano@ior.usi.ch; 3Instituto de Productos Naturales y Agrobiología (IPNA-CSIC), La Laguna, 38206 Tenerife, Spain; mcueto@ipna.csic.es; 4Instituto Universitario de Bio-Orgánica Antonio González (IUBO AG), Universidad de La Laguna (ULL), La Laguna, 38200 Tenerife, Spain; adiazmar@ull.edu.es; 5Department of Bioengineering, Faculty of Engineering, Manisa Celal Bayar University, 45119 Manisa, Turkey; iremdenz@gmail.com; 6Research Center for Testing and Analysis (EGE MATAL), Ege University Application, 35100 İzmir, Turkey; aysegul_erdogan@live.com; 7Department of Biology, Faculty of Science, University of Sarajevo, 71000 Sarajevo, Bosnia and Herzegovina; llbilela@pmf.unsa.ba; 8Marine Natural Products Team, UMR 7272, Institut de Chimie de Nice, Université Côte d’Azur, CNRS, 06108 Nice, France; corentin.moulin@univ-cotedazur.fr (C.M.); elisabeth.taffin-de-givenchy@univ-cotedazur.fr (E.T.-d.-G.); mehiri@unice.fr (M.M.); 9Faculty of Biomedical Sciences, USI, 6900 Lugano, Switzerland; giuseppe.perale@usi.ch; 10Ludwig Boltzmann Institute for Experimental and Clinical Traumatology, 1200 Vienna, Austria; 11Marine Biology Station Piran, National Institute of Biology, 1000 Ljubljana, Slovenia; ana.rotter@nib.si; 12Marine Biodiscovery Laboratory, School of Chemistry and Ryan Institute, National University of Ireland, Galway (NUI Galway), H91TK33 Galway, Ireland; olivier.thomas@nuigalway.ie; 13UCIBIO—Applied Biomolecular Sciences Unit, Department of Chemistry, Blue Biotechnology & Biomedicine Lab, Faculty of Science and Technology, NOVA University of Lisbon, 2829-516 Caparica, Portugal; 14Oncology Institute of Southern Switzerland, 6500 Bellinzona, Switzerland

**Keywords:** marine natural products, marine drugs, anticancer, drug discovery, clinical pipeline

## Abstract

The marine environment is a rich source of biologically active molecules for the treatment of human diseases, especially cancer. The adaptation to unique environmental conditions led marine organisms to evolve different pathways than their terrestrial counterparts, thus producing unique chemicals with a broad diversity and complexity. So far, more than 36,000 compounds have been isolated from marine micro- and macro-organisms including but not limited to fungi, bacteria, microalgae, macroalgae, sponges, corals, mollusks and tunicates, with hundreds of new marine natural products (MNPs) being discovered every year. Marine-based pharmaceuticals have started to impact modern pharmacology and different anti-cancer drugs derived from marine compounds have been approved for clinical use, such as: cytarabine, vidarabine, nelarabine (prodrug of ara-G), fludarabine phosphate (pro-drug of ara-A), trabectedin, eribulin mesylate, brentuximab vedotin, polatuzumab vedotin, enfortumab vedotin, belantamab mafodotin, plitidepsin, and lurbinectedin. This review focuses on the bioactive molecules derived from the marine environment with anticancer activity, discussing their families, origin, structural features and therapeutic use.

## 1. Introduction

Present in cells/tissues of microorganisms, fungi, plants or animals from both aquatic and terrestrial environments, natural products (NP, also called secondary or specialized metabolites) have been used for therapeutic purposes since ancient times and are important pillars of modern pharmacology [1,2,3,4,5,6,7,8]. Although the term secondary metabolite originates from the initial observation that their presence is not necessary for the vegetative growth, reproduction and development of organisms, they increase competitiveness with other species and play a fundamental role in the stress response, the defense against predators, the prevention of overgrowth by fouling organisms as well as the protection against bacterial, viral or fungal invasion, and UV radiation [1,7,8]. NPs are considered essential to overcoming stress conditions caused by fluctuating or changing environmental factors such as light intensity, temperature, humidity, salinity or mechanical wounding [1,7,8,9]. NPs play a crucial role in the survival of the organisms, functioning as modulators of complex interspecies interaction, where the boundaries between different types of interactions are not well defined nor always quite clear (e.g., different forms of symbiosis: parasitism, mutualism and commensalism) [10]. Thus, the mechanisms of survival and adaptation to environmental factors promote the production of highly complex and heterogeneous chemicals that have enormous potential to become drugs and cannot be matched by synthetic small molecules. In marine habitats, the high frequency of bioactive metabolites with toxic or deterrent effect is found in complex habitats such as coral reefs characterized by intense competition for space and by feeding pressure. Understanding cooperative and synergistic interactions, rather than competitive, has been significantly attenuated by studies of quorum sensing, cross-talk and electron transfer in microbial communities such as biofilms, which ensure more efficient survival strategies, particularly in extreme habitats [11,12]. Quorum sensing has been recognized as an important regulatory mechanism of secondary metabolite biosynthesis and/or cell differentiation in many bacteria, particularly streptomycetes. Finally, combined transcriptomic and metabolomics analyses highlighted the importance of exploring microbial species interactions to discover novel secondary metabolites with a wide range of biological properties [13].

Examples of terrestrial natural products that profoundly impacted drug discovery are: the salicylic acid from the bark of the willow tree used by ancient Egyptians and Greeks as a pain reliever [14] and precursor of aspirin, the most used nonsteroidal anti-inflammatory drug (NSAID) whose therapeutic effect is due to inhibition of the enzyme cyclo-oxygenase COX-2 [15]; the highly potent analgesic morphine, isolated from opium [16], that acts as an agonist of mu-opioid receptors inhibiting presynaptic and postsynaptic processes in the ascending pain transmission system [17]; the cardiotonic digitoxin, a secondary glycoside extracted from *Digitalis purpurea* [18], whose inhibition of the Na^+^/K^+^ ATPase membrane pump increases intracellular sodium and calcium concentrations and promotes the activation of contractile proteins of the heart [19]; penicillin, obtained from green mold *Penicillium notatum* in 1928, still one of the most widely used antibiotic agents [20], which binds through its four-membered beta-lactam ring the bacterial DD-transpeptidase enzyme, interfering with the peptidoglycan cell wall synthesis [21]. More recent examples of natural products that have impacted medicine, and in particular oncology, are the taxane paclitaxel, first isolated from the bark of the Pacific yew tree *Taxus brevifolia*, and the vinca alkaloids (vincristine and vinblastine), obtained from the Madagascar periwinkle plant *Catharanthus roseus*. They alter the dynamic equilibrium of polymerization or depolymerization of microtubules, compromising the mitotic spindle apparatus and causing apoptosis after metaphase arrest [22]. Camptothecin, extracted from the *Camptotheca acuminata* tree [23], is effective against a broad spectrum of tumors by blocking DNA topoisomerase I (topo I) [24]. To date, more than half of all drugs in clinical use have natural origin, and the percentage is higher for anti-cancer drugs [25,26,27]. The major classes of drugs developed from natural sources and used in clinics, along with the organism of origin, are reported in Figure 1.

The advances in technologies including “omics” tools for improved biodiscovery and the uncovering of metabolic pipelines, robotics, microfluidics, quantum computing, profiling, analytical and computational biology techniques for fractionation and isolation of secondary metabolites from crude extracts have considerably increased the exploration of natural sources. The integration of “omics” approaches with focus on genomics, transcriptomics and metabolomics, elucidates the complexity of gene regulatory networks and significantly helps our understanding of complex mechanisms involved in the expression of biosynthetic gene clusters encoding secondary metabolites [28].

Despite the success of terrestrial sources for the development of new drugs, our review emphasizes on those of marine origin due to their promising development.

Marine natural products are the result of enzymatic processes, and they are endowed by structural three-dimensional features which facilitate the binding to active sites, giving great specificity and unique bioactivity. Compared to synthetic drugs, they have more sp^3^-hybridized carbons, chiral centers, condensed aliphatic rings, fewer aromatic rings, higher number of carbon, hydrogen and oxygen atoms and more nitrogen content [3,29]. In addition, the presence of halogen atoms in their structures, mainly chlorine and bromine, is a result of their concentration in the surrounding seawater and specialized halogenating enzymes in organisms [30,31]. Natural products are usually distributed among different structural classes such as nucleosides, polyphenols, polysaccharides, peptides, alkaloids, polyketides (including macrolides), diketopiperazines, terpenoids (including steroids) and mixed biogenesis metabolites.

So far, the marine environment has been less investigated than the terrestrial one, but considering the differences of light, temperature, pressure and salinity of both environments and the vast biodiversity of the oceans, it is estimated that there will be a significant increase in the number of new molecules from marine origin added to the anticancer clinical pipeline. Indeed, recent decades saw a confirmation of this trend: today, there are already over 10 approved drugs for the treatment of patients affected by different types of cancer that derived from molecules available in the marine environment: cytarabine, nelarabine (prodrug of ara-G), fludarabine phosphate (pro-drug of ara-A), plitidepsin, midostaurin, eribulin mesylate, brentuximab vedotin, polatuzumab vedotin, enfortumab vedotin, belantamab mafodotin, trabectedin and lurbinectedin (Table 1). Moreover, others have moved through the early clinical evaluation and are now being evaluated in phase III of clinical trials for oncological indications (Table 2).

### Marine Environnent

The biological and chemical diversity of the marine environment is immeasurable and therefore represents an extraordinary resource for the discovery of new bioactive substances, drugs, toxins, pigments and enzymes, as well as biopolymers, bioadhesives, bioelastomers and hierarchically structured biocomposites [55]. Oceans represent 95% of the earth’s biosphere and over 70% of the planet’s surface [56]. Despite having numerous ecosystems and harboring the majority of the planet’s biodiversity, they are still largely unexplored. Less than 5% of the deep-sea has been investigated and an even lower percentage (0.01%) of the deep-sea floor has been sampled and studied [56]. In the marine environment thrive microorganisms (bacteria, actinobacteria, cyanobacteria, autotrophic flagellates, and diatoms, microalgae and fungi etc.), macroalgae (seaweeds), invertebrates (sponges, corals, worms, mollusks, echinoderms etc.) and flowering plants (seagrasses, mangroves and other halophytes etc.), for a total of species that may approach one to two million, with nearly 250,000 already described [57,58,59]. An overview of the species discovery rate through the decades is reported in Figure 2. Advances in the biodiscovery process have significantly increased the number of known marine species and their compounds. Accordingly, since 2008, more than 1000 new marine-derived compounds have been discovered each year [60].

The high marine biodiversity is the consequence of the wide range of unique conditions in which organisms survive. Temperatures fluctuate from −1.5 °C in the Antarctic waters to 350 °C in the hydrothermal systems. In the ocean, pressure increases by about one atmosphere for every 10 m, hence it can be 1000 times higher than the pressure at sea level [61]. Nutrient variations range from oligotrophic (low nutrient content) to eutrophic (nutrient-rich zones). Light intensity rapidly dissipates as depth increases, and there is no significant luminosity beyond 200 m. The near-surface photic zones receive sufficient sunlight to allow photosynthesis by marine plants and plankton. The deepest and more extensive non-photic zones are not illuminated enough to perform photosynthesis, hence other physical, chemical and biological processes such as bioluminescence have been developed to compensate for the lack of light. Similarly, the amount of free oxygen that is dissolved in the water varies from 7 mg/L at the surface to less than 0.1 mg/L in oxygen depleted zones depending on temperature, latitude, salinity, and depth [62,63]. The pH is generally around 8 and the salinity around 35 g/kg [64]. This extraordinary variety of conditions in marine habitats has induced species to develop mechanisms of evolution and adaptation not necessary in the terrestrial organisms. Therefore, marine-derived chemicals usually have unique structural scaffolds, biological modes of actions and have an important role as hit or lead compounds in drug discovery [7,65].

The geographic origin of the marine compounds reported since 2008 shows that almost a quarter of these are from Australia (24%), followed by a marked increase in metabolites discovered from the South China Sea (18%) and from the Pacific Ocean (17%). However, these data are not an indication of the geographical distribution of bioactive compounds in the seas but a mere consequence of funding investment in marine natural product chemistry research from Australia, China, South Korea, Japan and USA, and the accessibility level of deeper water to submersibles and remotely operated vehicles [66]. In Europe, marine biodiscovery has been stimulated since the fourth Framework Programme in 1994, but more intensively since 2007, based on the CORDIS database, the European Commission’s primary source of information on the projects funded by the EU Framework Programmes. While initially, projects focused on uncovering the marine biodiversity and function, in recent years the trend of funded projects has shifted to link marine bioprospecting with biotechnological applications.

## 2. Natural Product Classes of Marine-Derived Anticancer Agents

The natural product structural classes of the anti-cancer drugs coming from the marine environment (Table 1, Table 2 and Table 3) are varied, and we will report the principal structural features of each class together with their most representative compounds, approved or in clinical evaluation.

### 2.1. Nucleoside Derivatives

The first contribution to the study of marine and marine-inspired products as biologically active compounds that served as leads for drug discovery was published in the early 1950s by Bergmann et al., who described the arabinonucleoside spongothymidine (ara-T), isolated from the Caribbean sponge *Tectitethya crypta* (a.k.a. *Cryptotethya crypta* de Laubenfels, 1949) [67,68]. The chemical structure of ara-T, together with the other sponge-derived nucleosides spongosine and spongouridine, inspired the synthesis of the first two marketed drugs from marine origin: cytarabine (anticancer cytosine arabinoside or arabinosyl cytosine, ara-C) approved by the U.S. Food and Drug Administration (FDA) in 1969 and vidarabine (antiviral adenine arabinoside or arabinofuranosyladenine, ara-A) approved in 1976 (Figure 3). Therefore, nucleosides had a pivotal role in the history of marine-derived drugs. It is interesting to note that cytarabine and vidarabine were synthesized in laboratory prior to their extraction from natural sources *Streptomyces gryseus*, the gorgonian *Eunicella cavolini* and the fermentation broth of *Streptomyces antibioticus* strains [69].

Marine nucleosides are glycosylamines composed of a nitrogen-containing heterocyclic nucleobase (purine or pyrimidine) and a modified 5-carbon sugar (arabinose rather than ribose or deoxyribose) bounded through a β-glycosidic linkage. The compounds inspired by the sponge *Tectitethya crypta* are called ara-nucleoside to emphasize the presence of the arabinose moiety. Their discovery led to the new therapeutic principle that nucleosides could be used as chemotherapeutics. Arabinonucleosides ara-A and ara-G inspired the anticancer pro-drugs fludarabine phosphate and nelarabine, both FDA approved in 2005.

**Cytarabine** is a pyrimidine nucleoside containing a cytosine base. It is made from 1-β-d-arabinofuranosyluracil by acylation of its hydroxyl groups and conversion of the carbonyl group in position 4 of the pyrimidine ring into a thiocarbonyl group. The subsequent replacement of the mercaptan group with an amino group and the contemporary hydrolysis of the acetyl groups gives the final pyrimidine structure, also known with the chemical name of 1-arabinofuranosylcytosine (Figure 3) [73]. Cytarabine is a prodrug, intracellularly converted by deoxycytidine and pyrimidine kinases to the corresponding triphosphate activated metabolite (arabinosylcytosine triphosphate, ara-CTP), which competes with the physiological building block of nucleic acids 2′-deoxycytidine for incorporation into DNA [74]. Cytarabine differs from its naturally occurring counterpart by a hydroxyl group in the 2′-β configuration of the sugar moiety. The resulting arabinose binds to the replicating DNA strands and inhibits chain initiation and elongation, leading to the fabrication of faulty DNA. Additionally, the interference with DNA/RNA polymerases and nucleotide reductase enzymes blocks the cell cycle in S phase (DNA synthesis) [75,76,77]. Hence, rapidly dividing cells, which require more DNA replication for mitosis, are the most affected. Since the mechanism of action of cytarabine mimics the structure of a natural metabolite but not its function, causing cellular damage, it is an antimetabolite agent. It was the first marine-derived drug used for the treatment of leukemia and it has been widely used for hematological cancers since 1969. A liposomal formulation of cytarabine allows the improvement the stability and rapid deamination of the compound, cytarabine, increasing the half-life, and providing a prolonged drug exposure to tumor cells also in the central nervous system [78,79,80].

Efforts in producing new bioactive ara-nucleosides led to the synthesis of two additional antineoplastic antimetabolites: nelarabine and fludarabine phosphate. The marine-inspired **nelarabine** is a water-soluble prodrug of the cytotoxic deoxyguanosine analogue 9-β-d-arabinofuranosyl guanine (ara-G) that shares the same mechanism of action of cytarabine. Nelarabine is initially demethoxylated by endogenous adenosine deaminase to ara-G, whose phosphorylation into arabinosylguanine nucleotide triphosphate (ara-GTP) by deoxyguanosine and deoxycytidine kinases creates the cytotoxic metabolite responsible for blocking DNA synthesis. In 2005, it was approved by the FDA for the treatment of patients with relapsed or refractory T-cell lymphoblastic leukemia and T-cell lymphoblastic lymphoma following at least two prior chemotherapy regimens [81]. Ongoing clinical studies are currently defining its role in combination with other chemotherapy agents in the front-line setting of leukemias and lymphomas.

**Fludarabine phosphate** is the phosphate salt of a purine arabino-nucleoside that has 2-fluoroadenine as a nucleobase. Like the previous analogues, it is a prodrug that must be intracellularly converted in the active form. It is initially dephosphorylated to 2-fluoro-ara-A in plasma and then transported into the cell where it is phosphorylated by deoxycytidine kinase to 2-fluoro-ara-ATP. The latter, once incorporated into DNA, acts as a DNA chain terminator [82,83,84]. Its therapeutic uses include the treatment of adult patients with chronic lymphocytic leukemia who have not responded to at least one standard alkylating-agent containing regimens and since 2010, in combination with cyclophosphamide and rituximab [85]. More than 500 clinical trials are currently ongoing with fludarabine in multidrug regimens for treating hematological malignancies.

### 2.2. Macrolide Derivatives

Another prominent class of marine-derived natural products for therapeutic use are macrolides, polyketides featuring a macrocyclic lactone ring (Figure 4). Furthermore, one or more sugars may be attached via glycosidic bonds to the macrocycle. The complexity of macrolides often implies a difficult structural elucidation, especially the determination of the absolute configuration at each chiral center, and the chemical synthesis of their intricate chemical features is a challenging area of research in medicinal chemistry [86].

Several marine macrolides have been isolated from sponges, algae and other marine invertebrates, characterized by diverse and interesting biological properties but they have usually been proven to be produced by their associated microbiota. Among them, the analogue of the marine polyether macrolide halichondrin B, **eribulin mesylate**, was approved in 2010 for the treatment of metastatic breast cancer [87]. The approval was based on the phase III EMBRACE trial that demonstrated an improved overall survival versus physician’s choice in patients with previously treated with anthracyclines and taxanes [88]. Since 2016, eribulin is also FDA-approved for the treatment of unresectable and metastatic liposarcoma, based on the results of phase III trials by comparing the compound against dacarbazine [89,90].

The natural product **halichondrin B** was initially isolated in 1986 from the Japanese marine sponge *Halichondria okadai* [91]. It was later identified in extracts of the other poriferan species of genera *Axinella*, *Phakellia* and *Lissodendoryx* but it remained difficult to secure sufficient amounts of sample, which hampered its clinical development [92,93,94]. During the completion of the synthesis of halichondrin B in 1992 [95], through a long linear sequence requiring 47 steps, an intermediate C1-C38 was identified as the main fragment exhibiting cytotoxic activity. Maintaining the right-hand macrolactone and omitting the side chain, a potent and simplified analogue of halichondrin B was therefore obtained (Halichondrin B analogue E7389). A primary amine was inserted in place of the removed half of the original molecule. Just like its natural precursor, eribulin has been found to be a strong microtubule inhibitor. It binds, with high affinity, the plus ends of each protofilament, preventing microtubule growth and resulting in G_2_/M phase arrest and apoptosis. Eribulin-induced mitotic blockade is irreversible [96]. However, its mechanism is distinct from other antitubulin agents because it does not affect either the shortening phase inducing disassembly such as vinca alkaloids or the growing phases, such as taxanes. Such a mechanism of action can be attributed to a different interaction with the target. Eribulin binds microtubules in a high affinity binding site, different from those of other antitubulin agents. Unlike vinca alkaloids or the inner lumen of microtubules, such as taxanes, that bind to both α- and β-subunits, eribulin binds to a single site of interdimer interface or to the β-tubulin subunit [97]. These variations in site and mode of action make eribulin a mechanistically unique inhibitor of microtubule dynamics, hence it is extensively studied for the treatment of patients with taxane-resistant cancers, and with other solid tumors.

Marine fauna is also a source of another class of polyoxygenated macrolide lactones named **bryostatins**. The base structure of all bryostatins is the bryopyran ring system, a 20-membered macrolactone ring with three tetrahydropyran rings (A, B and C). Each ring is connected to the other with a different chain: a methylene bridge between A and B rings, a propenyl chain between B and C rings and a lactone substructure (ethylbutanoate) between C and A rings. Geminal methyl groups are also linked to the propenyl portion and the pyran ring A. Most bryostatins have an exocyclic methyl enoate in their B and C rings, others, instead, have a butenolide bound to the C-ring or glycals in place of hydroxyl moieties [98]. These compounds are produced by the proteobacterium *Candidatus Endobugula sertula* that lives in association with the marine bryozoan *Bugula neritina*. These compounds were found to protect the larvae of *Bugula neritina* from predation [99,100]. A direct evidence of this symbiosis is the reduction in the amount of bryostatins in the *B. neritina* colonies after antibiotic treatment [101,102].

The first bryostatin to be reported was bryostatin 3 in 1970 by Pettit et al. [103], followed by the isolation and purification of **bryostatin 1** in 1982 and many analogues over the years, reaching a current total of around 20 bryostatin analogues [104,105,106,107]. Structure–activity relationship studies have shown that C-1, C-19 and the oxygen in position C-26 are involved in the interaction with the target [108]. The mechanism of action of bryostatin 1 is linked to its ability to induce membrane translocation and activation of protein kinase Cs (PKCs) through the regulatory C1 domains. The modulation of PKC alters the expression of surface antigens in cancer cells thus increasing their susceptibility to immune or other antigen-targeted treatment strategies [109,110]. Among the components of this class, bryostatin 1 has been the focus of many preclinical and clinical studies [111,112,113,114,115,116,117,118,119,120,121]. Unfortunately, the scarce supply of this molecule from the natural source made its clinical development difficult. Thirteen tons of *B. neritina* were harvested to yield 18 g of bryostatin 1 and the proteobacterium seemed to be refractory to in vitro culture, adding difficulty to production through direct fermentation. Recently, a method for the total synthesis of bryostatin 1 has been developed [122], but it is not feasible for a large-scale production, launching the design of simpler analogues of bryostatin, known as “bryologues”.

### 2.3. Peptide Derivatives

Several types of peptides have been isolated from a wide variety of marine organisms and have been reported to have a broad range of bioactive properties, including antitumor [123,124]. Here also most of the studies so far have demonstrated a microbial origin for the production of these compounds. Extensive research has been conducted marine peptides and their potential applications continue to be unceasingly investigated so that the most recently approved marine-derived drugs belong to this class. It is estimated that in the last 10 years about 2500 new peptides with antitumor activity have been identified from natural sources [125].

They vary considerably in size and structural complexity and can be classified as linear depsipeptides, cyclic depsipeptides or linear peptides. Compared to linear peptides composed of a linear chain of amino acids linked by amide bonds, depsipetides have a more complex structure with one or more of the amide bonds replaced by the corresponding ester bonds (Figure 5) and a mode of action usually different from most of common peptides. Many of them contain atypical amino acids with post-translational modifications (carbamoylation, *N*- and *O*-methylations) or amino acids not found in common proteins.

A turning point in the development of marine-derived peptides for cancer chemotherapy was the discovery of **dolastatin 10**, a linear pentapeptide isolated in 1987 from the Indian Ocean sea hare *Dolabella auricularia*. Later this natural product was identified in the cyanobacteria *Symploca hynoides* and *Lyngbya majuscula* involved in the hare’s diet. Dolastatin 10 is composed of four amino acid building blocks, dolavaline (Dov), valine (Val), dolaisoleucine (Dil) and dolaproline (Dap) plus the C-terminal amine dolaphenine (Doe). It is the most potent member of a larger class of related peptides extracted from the sea hare, generically called dolastatins [126,127].

Preclinical studies reported that dolastatin 10 prevents tubulin polymerization similarly to vinca alkaloids, thus strongly affecting the microtubule equilibrium and leading to apoptosis. However, despite the encouraging in vitro activity, dolastatin 10 failed to progress to clinic trials due to its non-appreciable therapeutic index and significant toxic side effects at the maximum tolerated dose [128,129,130,131]. To exploit the potent antimitotic effect and improve the pharmacokinetic profile, structural modifications of dolastatin 10 have been developed, resulting in the successful class of auristatins, which present differences in the C-terminus, from dolaphenine (Doe) in dolastatin 10 to phenylalanine in monomethyl auristatin-F (Phe). More importantly, auristatins appeared as great payloads in antibody–drug conjugates (ADCs) [132]. The efficacy of ADCs to exploit antibodies to target specific molecules on tumor cells and selectively deliver otherwise highly toxic molecules [133] is evident by the fact that four compounds belonging to this class have recently been approved for the treatment of patients with cancer (Table 1) and over 20 other compounds are in clinical evaluation [134,135,136].

**Brentuximab vedotin** was the first commercially available ADC incorporating a payload derived from marine natural compounds: an average of four molecules of **monomethyl auristatin E (MMAE)** are linked to the anti-CD30 antibody cAC10 via the spacer para-aminobenzylcarbamate, a cathepsin cleavable linker (valine-citrulline) and an attachment group consisting of caproic acid and maleimide [39]. MMAE maintains the same structure of dolastatin 10 except for the *C*-terminal where the dolaphenine residue is substituted by (1*S*,2*R*)-(+)-norephedrine (Figure 5) [137]. Its high toxicity prevented MMAE from being used alone [138]. Therefore, the conjugation with an antibody is fundamental for its administration to combat cancer cells. Once internalized by clathrin-mediated endocytosis, lysosome proteases cleave the linker and releases MMAE into the cytosol. As its original molecule dolastatin 10, MMAE binds microtubules and potently inhibits polymerization, inducing G2/M phase cell cycle arrest and induce apoptosis. Compared to other antimitotic agents used for lymphomas such as vinblastine, this formulation shows potency of up to 200 times. Brentuximab vedotin is FDA-approved for different indications in Hodgkin’s lymphoma, systemic anaplastic large cell lymphoma, cutaneous and peripheral T-cell lymphomas [139,140,141,142].

**Polatuzumab vedotin** is another marketed ADC containing MMAE as payload [143], approved by the FDA in 2019 in combination with bendamustine and rituximab for the treatment of relapsed or refractory diffuse large B-cell lymphoma after at least two prior therapies [144]. An average of 3.5 MMAE molecules are conjugated to a humanized anti-CD79B IgG1 antibody [38]. The antibody’s target is highly expressed in B cells of lymphoma patients, making the therapy highly specific. The approval was based on a multicenter phase Ib/II study in relapsed or refractory diffuse large B-cell lymphoma patients considered transplantation-ineligible and who experienced treatment failure with prior ASCT [145].

MMAE is also the payload in **enfortumab vedotin,** approved by the FDA in 2019 for the treatment of patients with locally advanced or metastatic urothelial cancer who had received a prior programmed cell death (PD-1) or programmed death ligand 1 (PD-L1) inhibitor and platinum-containing chemotherapy regimen [146]. Here, the antibody targets Nectin-4, a cell surface protein involved in Ca^2+^-independent cellular adhesion, expressed in urothelial cancer cells [37]. The approval was based on the results of a phase II trial using enfortumab vedotin as monotherapy [147].

Besides these already approved compounds, MMAE has been employed to develop further compounds that have entered the clinical evaluation, including the anti-HER2 ALT-P7 [44] and disitamab vedotin [47] that have received the FDA orphan drug designation for gastric cancer.

Another synthetic auristatin derivative used as payload is **monomethyl auristatin F (MMAF)** that differs from MMAE for a phenylalanine at its C-terminus. This moiety attenuates the cytotoxic activity and contributes to membrane impermeability [148]. MMAF is incorporated in the anti-BCMA **belantamab mafodotin** [36], and was FDA approved in August 2020 for the treatment of patients with relapsed or refractory multiple myeloma based on a phase II study [149]. Other MMAF-based ADCs are undergoing clinical evaluation, including the anti-EGFR **depatuxizumab mafodotin** [54], which received the FDA orphan drug designation for glioblastoma.

In 2018, based on the randomized phase III ADMYRE trial, the new marine-derived anticancer peptide **plitidepsin** was approved by the Australian Therapeutic Goods Administration for the treatment of refractory multiple myeloma in combination with dexamethasone after at least three prior regimens [150,151]. Plitidepsin (or dehydrodidemnin B) is a cyclic depsipeptide isolated from the Mediterranean tunicate *Aplidium albicans* in 1991 and it belongs to the didemnin family [152,153]. The first congener of this class, didemnin B, was isolated from the Caribbean tunicate *Trididemnum solidum* in 1981. Because of poor efficacy and many severe side effects such as anaphylactic reactions and neuromuscular toxicity in phase-I and II trials, its development was abandoned in favor of the second generation didemnin, plitidepsin, more active and better tolerated in clinical studies [154]. Plitidepsin shows in vitro and in vivo anti-tumor activity [35,155,156]. Interestingly, the two marine species that contained didemnin B and plitidepsin were found in distant and remote geographic locations. The chemical structure of plitidespin closely resembles didemnin B but differs in the hydroxy group of the 1-(2-hydroxypropanoyl)-l-prolinamide moiety that has been oxidized to the corresponding ketone (Figure 5). The difficulties in harvesting *Aplidium albicans* and the lack of possible aquaculture alternative limits the natural supply of plitidepsin. Today it is manufactured by multi-step total synthesis, adding the three aminoacids (*R*)-*N*-Me-Leu linked to the Thr and piruvil-l-Pro to a backbone of six aminoacid subunits [157]. Plitidepsin antitumor activity appears to be related to the interaction with the eukaryotic Elongation Factor 1 (eEF1A2) protein [158], with induction of early oxidative stress, activation of Rac1 GTPase and suppression of protein phosphatases, that contribute to a rapid and persistent activation of both c-Jun *N*-terminal kinase (JNK) and p38 mitogen-activated protein kinases (p38/MAPK), and apoptosis induction [159]. In addition, it was demonstrated that protein kinase C delta (PKC-delta) mediates the cytotoxic effect of plitidepsin and is involved in caspase cascade activation and execution of apoptosis. In fact, cells deficient in PKC-delta have enhanced survival compared to wild type counterparts after drug treatment [159].

### 2.4. Alkaloids

Numerous amines and cyclic nitrogen-containing compounds named alkaloids are found in natural sources and their anticancer activities have been widely studied, essentially in terrestrial plants. Approximately two thousand alkaloids have been identified in terrestrial plants but more rarely in marine organisms. **Trabectedin** is a tetrahydroisoquinoline alkaloid, extracted from the Caribbean sea squirt *Ecteinascidia turbinata* during the extensive natural product isolation and screening program at the National Cancer Institute (NCI) in the 1960s [160], and later identified as a product of the bacterial symbiont *Candidatus Endoecteinascidia frumentensis* [161]. It is a very potent member of the ecteinascidin family of biologically active compounds containing two to three tetrahydroisoquinoline (THIQ) subunits and an active carbinolamine functional group. Trabectedin structure is characterized by three fused THIQ rings (A, B and C). The A and B rings form a rigid pentacyclic skeleton linked to ring C through a 10-membered lactone bridge (Figure 6). The intricacy of this structure took over 20 years from the detection of its activity to being fully elucidated [162]. Ring A and B promote the alkylation to vicinal nucleotides in the same or opposite strands of the DNA minor groove through hydrogen bonds and Van der Waals interactions, creating a drug–DNA adduct responsible for trabectedin’s anticancer activity. The N2 of middle guanine in the 5′-CGG, TGG, GGC, AGC sequences are the sites of alkylation [163]. In addition, the protonated amine acts as a catalyst in the generation of the active iminium ion (N2) involved in the DNA binding [164]. The C ring protrudes from the DNA backbone and interferes with the function of DNA-binding proteins such as transcription factors and transcription-coupled nucleotide excision repair process, which in normal condition repairs the DNA damage caused by UV, cisplatin or other anticancer agents. This hypothesis is plausible since cell lines deficient in nucleotide excision repair proteins were less sensitive to trabectedin [165,166].

As for many others marine compounds, the low yield was a relevant problem in trabectedin clinical development (1 g of trabectedin is isolated from 1 ton of sea squirt). This was solved by exploiting a semisynthetic method from the microbial product cyanosafracin B, an antibiotic obtained by fermentation of the bacterium *Pseudomonas fluorescens* [167]. The anticancer activity of trabectedin is supported by data obtained in different tumor models, and its mechanism of action also involved an effect on the tumor microenvironment, mainly on macrophages [3,168,169,170,171,172,173,174,175,176,177,178]. Since 2007, trabectedin is approved by the EMEA for the treatment of patients with advanced soft tissue sarcoma, after failure of anthracyclines and ifosfamide or for those who are unsuited to receive it. In October 2015, the FDA approved trabectedin for the treatment of patients with unresectable or metastatic liposarcoma and leiomyosarcoma who received a previous anthracycline-based regimen. This new indication was based on a phase III study that showed a 45% reduction in the risk of disease progression or death compared to dacarbazine and made trabectedin the first drug to receive FDA approval for the treatment of liposarcoma [179]. Currently, trabectedin is in clinical evaluation for other cancer types, including breast, bone, prostate and ovarian cancer, both alone or in multidrug regimens.

**Lurbinectedin** is a synthetic tetrahydropyrrolo [4, 3, 2-de]quinolin-8(1H)-one analogue of the marine compound ET-736 isolated from *E. turbinata* in which the pentacyclic skeleton made of ring A and B is maintained while the ring C is substituted by a tetrahydro-β-carboline (Figure 6).

In June 2020, lurbinectedin was FDA approved for adult patients with metastatic small cell lung cancer with disease progression on or after platinum-based chemotherapy. This was based on the results of a phase II study reporting an ORR of 35% with a median response duration of 5.3 months [180]. While, in the CORAIL phase III trial, lurbinectedin was not superior to pegylated liposomal doxorubicin or topotecan in platinum resistant ovarian cancer patients [181], results are still expected for a phase III trial comparing lurbinectedin plus doxorubicin versus cyclophosphamide, doxorubicin and vincristine (CAV) or topotecan as treatment in small cell lung cancer patients who failed one prior platinum-containing line (NCT02566993).

Trabectedin and lurbinectedin have multiple mechanisms of action [182,183,184,185,186]. They bind minor groove CG-rich sequences located at promoters of protein-coding genes, interacting transcription factors or DNA repair molecules, inducing cell cycle perturbations and cell death [184]. Both compounds also decrease the number of tumour-associated macrophages, thus modulating the tumour micro-environment.

Staurosporine is an indolocarbazole alkaloid isolated in 1977 from *Streptomyces staurosporeus*, a terrestrial bacterium but later on also from certain marine species such as the sea squirt *Eudistoma toealensis* and the plathelminth *Pseudoceros* sp [187,188,189]. Although this molecule is very toxic, analogues were developed, such as **midostaurin** [190]. The latter is an FDA approved kinase inhibitor for the treatment of adult patients with newly diagnosed, FMS-like tyrosine kinase 3 (FLT3) mutation-positive acute myeloid leukemia in combination with standard cytarabine and daunorubicin induction, and cytarabine consolidation, or with aggressive systemic mastocytosis, systemic mastocytosis with associated hematological neoplasm or mast cell leukemia. Midostaurin targets several protein kinases such as FLT3 (Fms-like tyrosine kinase 3), PKC (protein kinases C) and VEGFRs (vascular endothelial growth factor receptors). Compared with the structure of staurosporine, the addition of a phenolic group increases the capacity of the molecule to be absorbed and decreases the toxicity of the molecule. Additional kinase inhibitors derived from marine compounds are the indolocarbazoles lestaurtinib (multikinase inhibitor with activity against FLT3, JAK-2, TRK-A, TRK-B, TRK-C), enzastaurin (PKCβ and GSK-3β inhibitor) and CEP-2563 (TRK-A/B/C inhibitor).

**Plinabulin** (Figure 6) is a synthetic dehydrodiketopiperazine, derivative of the diketopiperazine alkaloid halimide, one of the most clinically successful metabolites originally isolated from the marine fungi *Aspergillus sp*. and *Aspergillus ustus*. Plinabulin is a tubulin polymerization inhibitor interacting with the colchicine-binding site of β-tubulin, thereby interrupting equilibrium of microtubule dynamics. It inhibits the growth of proliferating vascular endothelial cells, disrupting the tumor vasculature that contributes to neoplastic proliferation [70,191]. Plinabulin might also act as immunomodulator and reduce chemotherapy-induced neutropenia and thrombocytopenia [192,193,194]. This compound is currently being evaluated in multiple clinical trials.

The marine-alkaloid **zalypsis** has been evaluated in both phase I and phase II trials for the treatment of solid tumors, lymphoma, multiple myeloma, Ewing’s sarcoma and primitive neuroectodermal tumor [195,196,197,198,199,200,201,202]. It is a synthetic tetrahydroisoquinoline alkaloid (Figure 6) related to jorumycin, a complex pentacyclic organic compound isolated from the mantle and mucus of the Pacific nudibranch mollusk *Jorunna funebris*. Zalypsis has demonstrated significant in vitro and in vivo activity against human solid and hematologic neoplasms [202,203]. Zalypis produces covalent modification of guanines in the DNA minor groove that give rise to DNA double-strand breaks, S-phase arrest and apoptosis in cancer cells [201].

### 2.5. β-Lactones

Actinobacteria and fungi are the major producers of β-lactone natural products, although they can be also isolated from higher plants and animals. β-Lactones are four-membered heterocycles with high reactivity. They are prone to react through nucleophilic attack with serine, threonine, or cysteine residues, therefore enzymes with these activated catalytic nucleophiles are susceptible to inhibition by the formation of stable covalent adducts with β-lactones. Relevant target of β-lactones are lipases and proteases [204]. So far, orlistat, a synthetic derivative of the natural product lipstatin, is the only β-lactone compound that has reached the pharmaceutical market. Lipstatin was isolated from extracts of a terrestrial *Streptomyces* strain and showed a potent inhibition of pancreatic lipase. Orlistat is the saturated derivative of lipstatin and is in the market under the name of Xenical as an anti-obesity drug.

The marine environment has proved to be a valuable source of Actinobacteria, and some of them are obligate marine bacteria due to their seawater requirement, such as members of the *Salinispora* genus. The family of compounds known as salinosporamides was isolated from marine Actinobacteria *Salinispora tropica* [205]. These compounds are characterized by the presence of a fused γ-lactam-β-lactone densely functionalized (Figure 7). The first member of this family was **marizomib** (salinosporamide A) and its structure presents a chloroethyl group at C-2, a methyl group at C-3 and a cyclohexene at C-5 [71,205]. This compound acts as a potent inhibitor of the 20*S* proteasome by covalently modifying its active site threonine residues. Marizomib entered clinical trial for the treatment of several types of cancer shortly after its publication in 2003 and at the present it is in phase III for the treatment of glioblastoma. It is interesting to point out that the marizomib supply for the clinical trials is obtained from the fermentation of *Salinispora tropica* after optimization of fermentation conditions [206]. There is a growing number of salinosporamide derivatives that have been obtain since the discovery of salinosporamide A. Most of them are natural products isolated from extracts of strains of the genus *Salinispora* and others are the result of chemical transformations of marizomib itself. However, so far, none of the new derivatives is able to improve the strong inhibition activity of marizomib against the 20S proteasome [207].

### 2.6. Polysaccharides

Besides phlorotannins, polysaccharides from marine macroalgae have been proposed as relevant adjuvant supplements for cancer management or used directly for the treatment [208]. Polysaccharides or glycans are large polymers composed of tens to thousands of monosaccharides like d-glucose, d-fructose, d-galactose, l-galactose, d-mannose, l-arabinose, and d-xylose joined through glycosidic linkages. Amino sugars (d-glucosamine and d-galactosamine), their derivatives (*N*-acetylneuraminic acid and *N*-acetylmuramic acid) or sugar acids (glucuronic and iduronic acids) are also frequently present [209]. In addition, seaweed-derived polysaccharides include agar, alginate and carrageenans as the main structural materials of cell walls and intercellular matrices. Agar is a complex mixture of two major polysaccharides: agarose that is a linear, neutral and low sulphate/methoxyl polymer constituting about 70% of the mixture plus agaropectin that is a charged, heterogeneous and highly-substituted fraction. It is extracted from red algae, primarily from the genera *Gelidium* and *Gracilaria.* Alginate is a linear polysaccharide composed of β-d-mannuronic acid (M) and α-l-guluronic acid (G), which aggregate to form a backbone with sequences of mannuronic acid blocks (M-blocks) or guluronic acid blocks (G-blocks) and regions of alternating residues. Alginates are structural components of the brown macro-algae Phaeophyceae and Rhodophyta. Carrageenans are linear, sulphated galactans made up of disaccharide repeating units: alternating 3-linked β-d-galactopyranose and 4-linked α-d-galactopyranose or 4-linked 3,6-anhydro-α-d-galactopyranos. Carrageenans are predominantly extracted from *Kappaphycus alvarezii*, *Eucheuma denticulatum*, *Gigartina skottsbergi* and *Sarcothalia crispate* red algae. Ι-carrageenan is FDA-approved as a thickener, stabilizer, and emulsifier in food manufacturing. Furthermore, it is currently in clinical trials for viral infections and allergic rhinitis.

The active macroalgal polysaccharides are principally the sulfated ones. Compared with other sulfated polysaccharides, fucoidans have been extensively investigated in recent years for drug development. The most therapeutically promising compound of this class is **fucoidan** (sulphated l-fucose) derived from the cell wall of various brown algae such as *Cladosiphon okamuranus, Fucus vesiculosus, Undaria pinnatifida* and *Sargassum fulvellum* (Figure 8). Many studies report its anticancer effect in various tumor cells [210,211,212,213,214,215] and fucoidan is undergoing clinical evaluation in phase II. The exact mechanism of action of fucoidan has not been defined yet, although data support both a direct and indirect anti-cancer activity, mediated at least in part by the inhibition of p38 MAPK and PI3K/AKT signaling pathways [216,217,218].

Among the sulfated polysaccharide that have shown anti-tumor activity in minor preclinical studies are: B-1, isolated from the culture of marine *Pseudomonas sp*. [219]; H3-a1, isolated from the brown seaweed *Hydroclathrus clathratus* [220]; the alkali-extracted polysaccharide DAEB from the green algae *Enteromorpha intestinalis* [221]; SargA from the brown algae *Sargassum stenophyllum* [222] and calcium spirulan (Ca-SP) from the blue-green alga *Spirulina platensis* [223]. Furthermore, sulfated glycosaminoglycans (GAGs) interfered with transcription functionality, inducing apoptosis in murine melanoma cells [224].

## 3. Conclusions

The diversity of biological materials of marine origin is almost equivalent to the marine biodiversity itself and the marine environment is an invaluable and underestimated source of anticancer compounds. After cytarabine FDA-approval in 1969, other marine-derived drugs have been approved for cancer treatment, with the past decade being the most productive in commercially exploiting marine-derived compounds. Importantly, drug development from marine sources is a time-consuming, high risk process and it requires high investment, entailing collaboration between scientific organizations, small and medium R&D companies and the pharma industry. An efficient marine drugs clinical pipeline thus necessarily involves the collaboration of (micro) biologists, chemists, toxicologists, medical and “omics” experts that jointly collaborate in the first stages, i.e., the organisms’ source, natural products discovery and preclinical trials. The initial stages of the pipeline are thus heavily dependent on the national and international scientific funding resources as well as open access to data from previous research activities. Medical, legal (including regulatory and intellectual property) as well as ethics experts are then needed to efficiently progress in the clinical trials stage, ideally leading to a development finalization and market entry in case of demonstrated potency, efficiency, safety and return on investment. This is a lengthy process, with limitations due to lack of sustainable supply, structural complexity, and poor pharmacokinetic properties that have been overcome by full synthetic manufacture, new procurement technologies including the optimization of fermentation processes as in the case of marizomib, the encapsulation of drugs in nanoparticles and, as a very successful approach, their use as payloads in ADCs. Hence, marine-derived compounds have an important and increasing role in the discovery of new anticancer drugs. With the advance of “omics” tools that, when combined aid in dereplication, structure elucidation and production, such as genomics (which also can aid absolute configuration assignment), metabolomics and enzymatic synthesis of bioactive metabolites, and the development of integrated MS/MS and NMR databases, the next decade looks very promising in terms of marine natural product discovery and drug development, advancing clinical pipeline with decreased times from hit to lead, for lead optimization, and lead to drug candidate.

## Figures and Tables

**Figure 1 marinedrugs-18-00619-f001:**
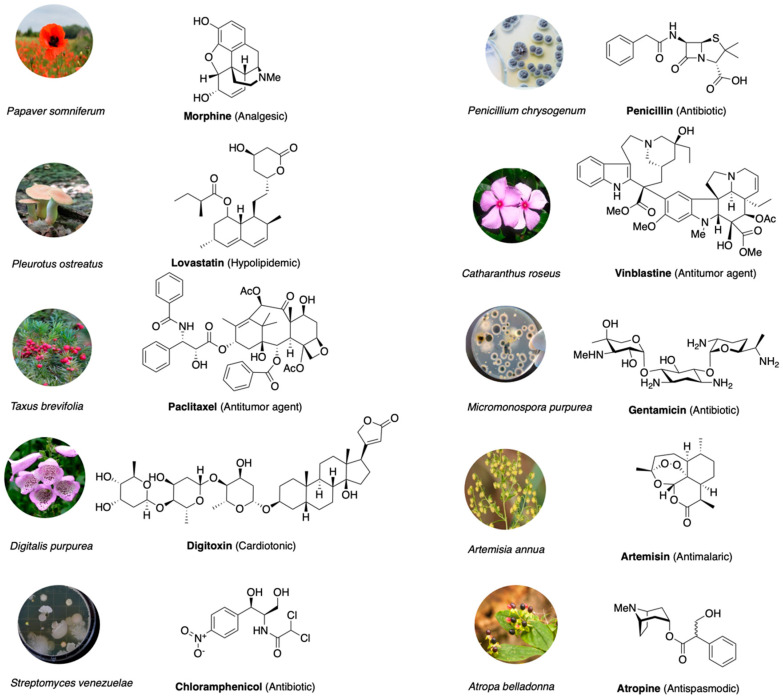
Drugs developed from natural sources, their biological sources, chemical structures and treatment usage.

**Figure 2 marinedrugs-18-00619-f002:**
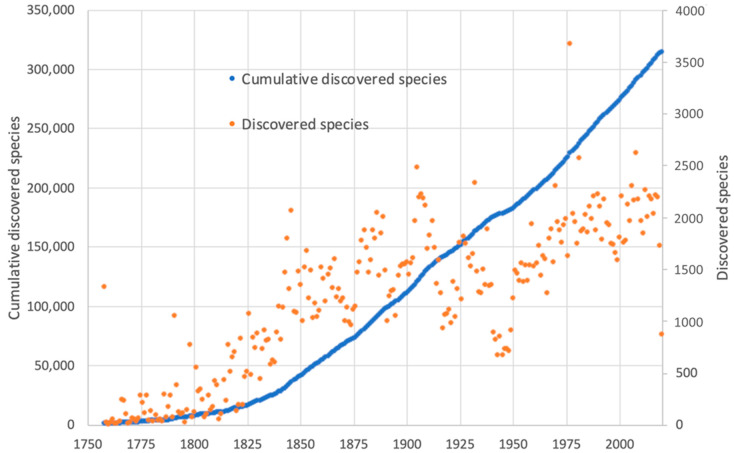
New species discovery rate, according to data available in the World Register of Marine Species. Orange and right Y-axis, number of discovered species; blue dots and left Y-axis, cumulative number of discovered species; X-axis, years. Data extracted from http://www.marinespecies.org, accessed in June 2020.

**Figure 3 marinedrugs-18-00619-f003:**
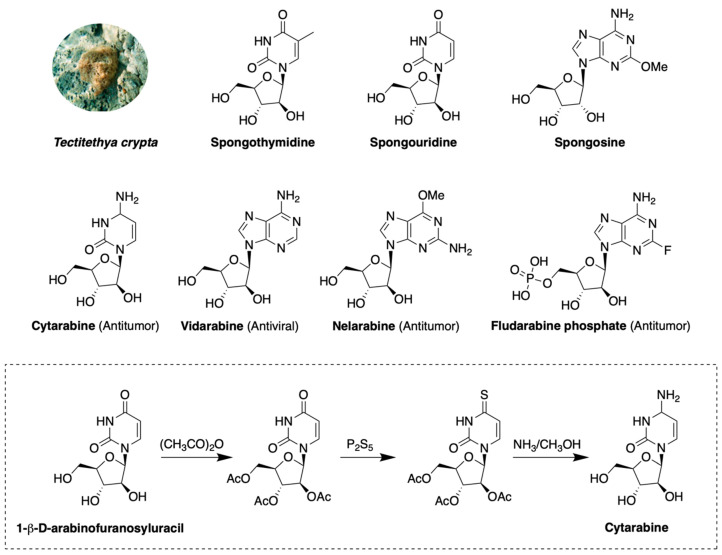
Ara-nucleosides structures and total cytarabine synthesis.

**Figure 4 marinedrugs-18-00619-f004:**
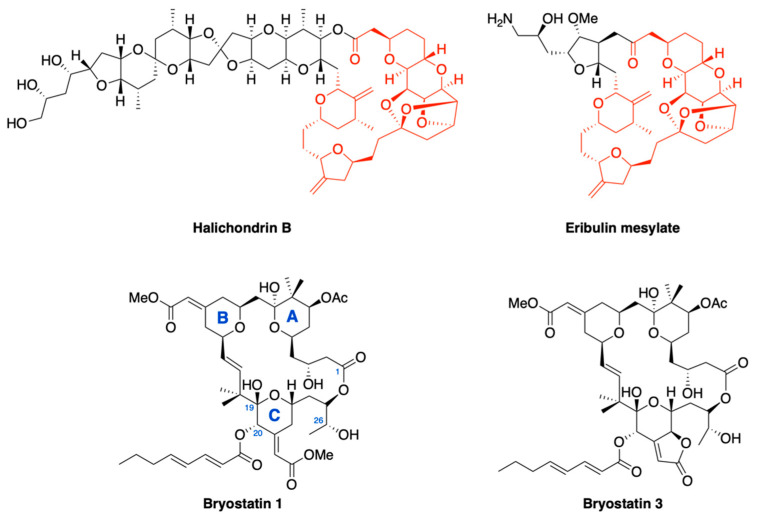
Marine-derived macrolides. The right macrolactone portion is essential to obtain halichondrin B-like activity. The C19–C20 region and the O26 of bryostatins are crucial in conferring biological response.

**Figure 5 marinedrugs-18-00619-f005:**
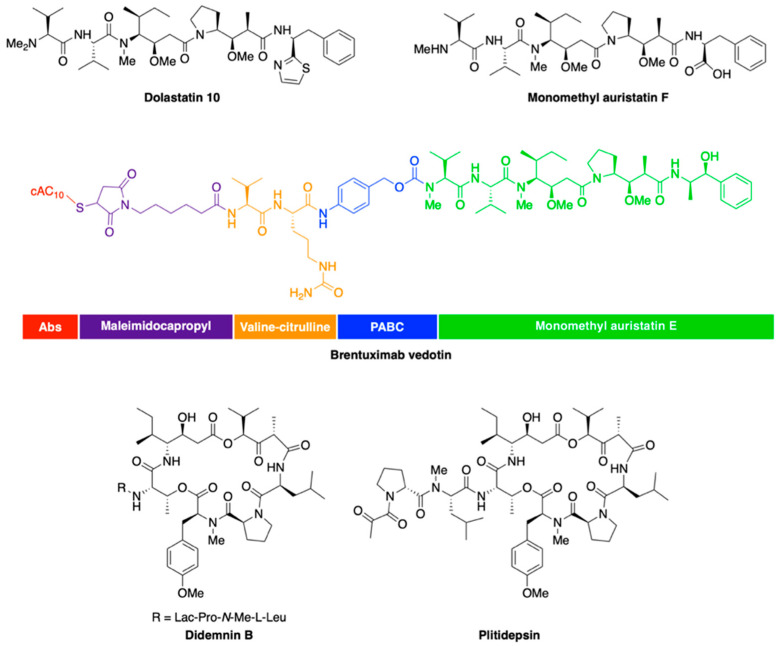
Marine peptides and their derivatives of pharmacological relevance.

**Figure 6 marinedrugs-18-00619-f006:**
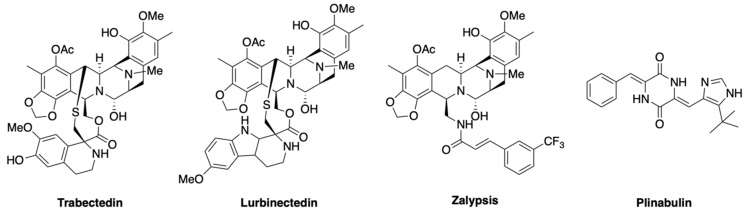
Marine alkaloids and their synthetic derivatives.

**Figure 7 marinedrugs-18-00619-f007:**
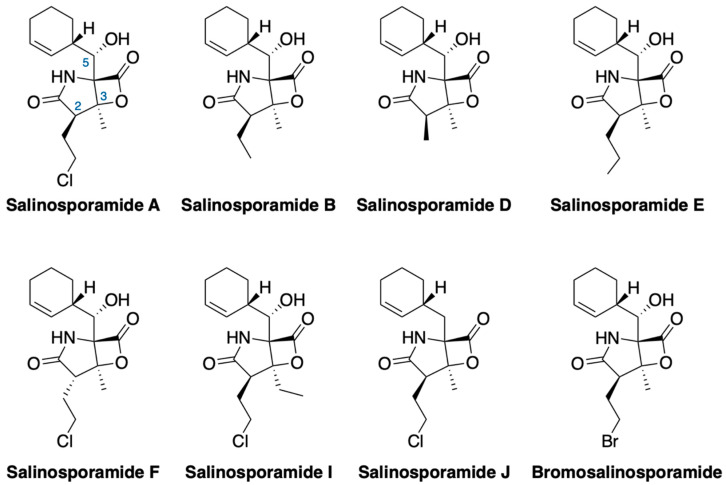
γ-lactam-β-lactone products of the salinosporamide family.

**Figure 8 marinedrugs-18-00619-f008:**
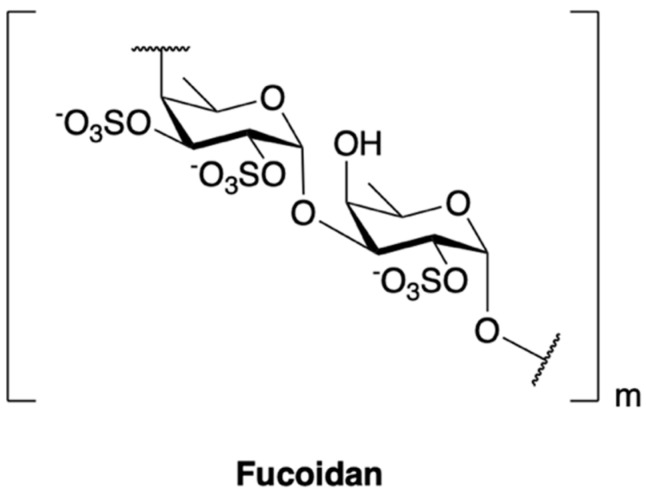
Fucoidan chemical structure is composed of a backbone of α-linked l-fucose residues with various substitutions including hydroxyl, methyl and sulfate ester groups.

**Table 1 marinedrugs-18-00619-t001:** Drugs approved as anti-cancer compounds and derived from marine compounds. Approved indications as assessed using http://adisinsight.springer.com/ in October 2020.

Compound	Marine Organism	Chemical Classes	Therapeutic Use	Ref.
Lurbinectedin	Tunicate	Alkaloid	Solid tumors	[32]
Trabectedin	Tunicate	Alkaloid	Solid tumors	[33]
Midostaurin	Tunicate/Actinobacteria	Indolocarbazole	Leukemias	[34]
Plitidepsin ^	Tunicate	Peptide	Multiple myeloma	[35]
Belantamab mafodotin	Mollusk/Cyanobacteria	ADC/Peptide **	Multiple myeloma	[36]
Enfortumab vedotin	Mollusk/Cyanobacteria	ADC/Peptide *	Solid tumors	[37]
Polatuzumab vedotin	Mollusk/Cyanobacteria	ADC/Peptide *	Lymphomas	[38]
Brentuximab vedotin	Mollusk/Cyanobacteria	ADC/Peptide *	Lymphomas	[39]
Eribulin mesylate	Sponge	Macrolide polyketide	Solid tumors	[40]
Fludarabine phosphate	Sponge	Nucleoside	Leukemias, lymphomas	[41]
Cytarabine	Sponge	Nucleoside	Leukemias, lymphomas	[42]
Nelarabine	Sponge	Nucleoside	Leukemias, lymphomas	[43]

ADC, antibody drug conjugate; ^, approved by the Australian Therapeutic Goods Administration; *, monomethyl auristatin E (MMAE) as payload; **, monomethyl auristatin F (MMAF).

**Table 2 marinedrugs-18-00619-t002:** ADCs containing MMAE or MMAF as payloads in clinical trials. Data assessed on https://clinicaltrials.gov (recruiting or active/not recruiting) in October 2020. Sorted by payload and by clinical status.

Clinical Status	Compound	Target	Payload	Marine Organism	Therapeutic Use	FDA Orphan Drug Designation	Ref.
Phase I	ALT-P7	HER2	MMAE	Mollusk/Cyanobacteria	Solid tumors	Gastric cancer	[44]
Phase I	RC88	Mesothelin	MMAE	Mollusk/Cyanobacteria	Solid tumors	-	*
Phase I	SGN-CD228A	CD228	MMAE	Mollusk/Cyanobacteria	Solid tumors	-	[45]
Phase II	CX-2029	CD71	MMAE	Mollusk/Cyanobacteria	Solid tumors, lymphomas	-	[46]
Phase II	Disitamab vedotin	HER2	MMAE	Mollusk/Cyanobacteria	Solid tumors	Gastric cancer	[47]
Phase II	Enapotamab vedotin	AXL	MMAE	Mollusk/Cyanobacteria	Solid tumors	-	[48]
Phase II	Ladiratuzumab vedotin	LIV-1 A	MMAE	Mollusk/Cyanobacteria	Solid tumors	-	[49]
Phase II	Telisotuzumab vedotin	MET	MMAE	Mollusk/Cyanobacteria	Solid tumors	-	[50]
Phase II	Tisotumab vedotin	TF	MMAE	Mollusk/Cyanobacteria	Solid tumors	-	[51]
Phase I	FS-1502	HER2	MMAF	Mollusk/Cyanobacteria	Solid tumors		[52]
Phase II	AGS 16C3F	ENPP3	MMAF	Mollusk/Cyanobacteria	Solid tumors	-	[53]
Phase III	Depatuxizumab mafodotin	EGFR	MMAF	Mollusk/Cyanobacteria	Solid tumors	Glioblastoma	[54]

FDA, Food and Drug Administration; MMAE monomethyl auristatin E; MMAF, monomethyl auristatin F; *, www.cancer.gov.

**Table 3 marinedrugs-18-00619-t003:** Marine compounds in clinical trials as anticancer agents. Data assessed on https://clinicaltrials.gov (recruiting or active/not recruiting) in October 2020.

Clinical Status	Compound	Marine Organism	Chemical Classes	Therapeutic Use	FDA Orphan Drug Designation	Ref.
Phase III	Plinabulin	Fungus	Alkaloid	Solid tumors	-	[70]
Phase III	Marizomib	Actinobacteria	Beta-lactone	Solid tumors, lymphomas, multiple myeloma	Multiple myeloma, glioblastoma	[71]
Phase II	Oligo-fucoidan	Brown seaweed	Polysaccharide	Solid tumors	-	[72]

FDA, Food and Drug Administration.

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
