# Peer review of "Marine Anticancer Agents: An Overview with a Particular Focus on Their Chemical Classes"

_marinedrugs, 2020, doi:10.3390/md18120619_

Round 1
Reviewer 1 Report
This is an interesting review focusing on anticancer agents from marine natural source, the review is well-written, However, the mechanism by which these anticancer agents used in cancer treatment in not discussed thoroughly.
The authors should address the following points:
The authors argue that the biological function of many (example- table 1) but how these agent target in specific should be rephrased in many results and discussion section.
The review paper would be strengthened by further discussion on recent development in the field (example this old decade story! Reference 34 “ Talley, R. W.; O'Bryan, R. M.; Tucker, W. G.; Loo, R. V., Clinical pharmacology and human antitumor 700 activity of cytosine arabinoside. Cancer 1967, 20, (5), 809-16.”) with some additional chemical classes. Not all of the experiments listed below would necessarily need to be done, but at least some of them seem necessary to bolster the interpretations.
I would recommend this review after minor revision with above suggestions.
Author Response
this is an interesting review focusing on anticancer agents from marine natural source, the review is well-written.
A:. We thank the Reviewer for the positive feedback
However, the mechanism by which these anticancer agents used in cancer treatment in not discussed thoroughly. The authors should address the following points:
The authors argue that the biological function of many (example- table 1) but how these agent target in specific should be rephrased in many results and discussion section.
A:. We agree with the Reviewer that the mechanisms of action had not been described for all the main compounds. We have now improved the text and especially filled the gaps regarding all the compounds included in figure 1, bryostatin 1, trabectedin and lurbinectedin, zalypis and fucoidan, which were indeed missing the information.
The review paper would be strengthened by further discussion on recent development in the field (example this old decade story! Reference 34 “ Talley, R. W.; O'Bryan, R. M.; Tucker, W. G.; Loo, R. V., Clinical pharmacology and human antitumor 700 activity of cytosine arabinoside. Cancer 1967, 20, (5), 809-16.”) with some additional chemical classes. Not all of the experiments listed below would necessarily need to be done, but at least some of them seem necessary to bolster the interpretations.
A:. We thank the Reviewer for the suggestion. We have improved the manuscript, albeit leaving some of the “vintage” references.
I would recommend this review after minor revision with above suggestions.
A:. We thank the Reviewer for the positive feedback
Reviewer 2 Report
In all aspects a concise and welcome review on marine products that are currently in use in various cancer therapies or are under investigation in phase two or three as potential compounds to be used in different cancer treatments. I request a minor review of the paper as follows:
line:
74 typo nteractions - interactions
80 interspecies - species
82 examples of terrestrial natural products
Fig 1 Pleutotus - Pleurotus
Fig 2 The expression "discovered species names" - in my opinion the species names cannot be discovered they can be given to a species.
309 oxygen position at 26 - at position 26 is hydroxyl group (-OH)
316 What are bryologues?
361 arrest cell cycle and induce apoptosis (according to the statement in line 342)
Fig 5 dolastatin and auristatin both have phenylalanine at C-terminus (which is not as stated in the text in line 383 and 384)
The last subchapter on polysaccharides is a bit out of the context. If such compounds are listed as potential anticancer compounds at least basic molecular mechanism should be revealed or targets in cancer cells should be mentioned. Otherwise the chapter is of not of particular merit.
Author Response
In all aspects a concise and welcome review on marine products that are currently in use in various cancer therapies or are under investigation in phase two or three as potential compounds to be used in different cancer treatments.
A:. We thank the Reviewer for the positive feedback
I request a minor review of the paper as follows:
line:
74 typo nteractions - interactions
A:. The text has been corrected.
80 interspecies - species
A:. The text has been corrected.
82 examples of terrestrial natural products
A:. The word has been added has been corrected.
Fig 1 Pleutotus – Pleurotus
A:. The figure has been corrected.
Fig 2 The expression "discovered species names" - in my opinion the species names cannot be discovered they can be given to a species.
A: The figure has been modified.
309 oxygen position at 26 - at position 26 is hydroxyl group (-OH)
A: We have corrected the text.
316 What are bryologues?
A: The text has been modified adding an explanation.
361 arrest cell cycle and induce apoptosis (according to the statement in line 342)
A:. The text has been corrected.
Fig 5 dolastatin and auristatin both have phenylalanine at C-terminus (which is not as stated in the text in line 383 and 384)
A:.Dolastatin 10 and monomethyl auristatin-D have a dolaphenine C-terminus, whereas auristatin-E (3) and monomethyl auristatin-E have a (1S,2R)-(+)-norephedrine. In the case of monomethyl auristatin-F, it does have a phenylalanine C-terminus. The manuscript has been corrected accordingly.
The last subchapter on polysaccharides is a bit out of the context. If such compounds are listed as potential anticancer compounds at least basic molecular mechanism should be revealed or targets in cancer cells should be mentioned. Otherwise the chapter is of not of particular merit.
A: We have modified the text, adding a brief introductory paragraph to the subchapter, which we would prefer to maintain in the manuscript. We have also added information of the mechanism of action of fucoidan as anti-cancer compound.
Reviewer 3 Report
The authors collected diffrent studies about the anticancer compounds which derived from the marine sources.
The topic of the review was discussed in different reviews and manuscripts before.
different points in teh review are missing the citation which is a copyright issue.
Author Response
The authors collected diffrent studies about the anticancer compounds which derived from the marine sources.
The topic of the review was discussed in different reviews and manuscripts before.
A:. We agree with the Reviewer that the topic is not novel but it is a relevant one and we believe our manuscript written by teams coming from different scientific backgrounds can add something to the field.
different points in teh review are missing the citation which is a copyright issue.
A: Sources had been properly cited. Following the Reviewer’s suggestions, we have better cited references, paragraph by paragraph.

Reviewer 4 Report
The authors have assembled a nice review on marine anticancer agents. I like that they focused on products that are either approved or moving through clinical trials. That level of focus provides a nice focus for the review, and I believe this will be of broad interest to the marine natural products community and the larger natural products community as well. Overall, the paper is well written and quite clear. It contains a nice level of detail. I only had a few minor suggestions as outline below:
- Table 2. Some of the columns need to be adjusted. For example, the payload column should be expanded so that all letters of the acronyms are on one line.
Line 151: There should be a period following "compounds" not a comma.
Line 163: A comma should be inserted after "increases."
Line 264: "has been approved" should be changed to "was approved." The same goes for line 391.
Line 322: "recent approved" should be "recently approved"
Line 352: This needs to be reworded. The first ADC using a natural product was mylotarg. Perhaps the authors just want to state that the first ADC that contained a marine natural product was brentuximab vedotin.
Author Response
The authors have assembled a nice review on marine anticancer agents. I like that they focused The authors have assembled a nice review on marine anticancer agents. I like that they focused on products that are either approved or moving through clinical trials. That level of focus provides a nice focus for the review, and I believe this will be of broad interest to the marine natural products community and the larger natural products community as well. Overall, the paper is well written and quite clear. It contains a nice level of detail.
A:. We thank the Reviewer for the positive feedback
I only had a few minor suggestions as outline below:
- Table 2. Some of the columns need to be adjusted. For example, the payload column should be expanded so that all letters of the acronyms are on one line.
A: The table has been resized.
Line 151: There should be a period following "compounds" not a comma.
A:. The text has been corrected.
Line 163: A comma should be inserted after "increases."
A:. The text has been corrected.
Line 264: "has been approved" should be changed to "was approved." The same goes for line 391.
A:. The text has been corrected in both locations.
Line 322: "recent approved" should be "recently approved"
A:. The text has been corrected.
Line 352: This needs to be reworded. The first ADC using a natural product was mylotarg. Perhaps the authors just want to state that the first ADC that contained a marine natural product was brentuximab vedotin.
A:. We agree with the Reviewer. The text has been corrected.
Round 2
Reviewer 3 Report
Thanks to the authors , they highly improved the manuscript by the addition of more refrences and machanisms of action.